# The Applications of 3D Imaging and Indocyanine Green Dye Fluorescence in Laparoscopic Liver Surgery

**DOI:** 10.3390/diagnostics11122169

**Published:** 2021-11-23

**Authors:** Giammauro Berardi, Marco Colasanti, Roberto Luca Meniconi, Stefano Ferretti, Nicola Guglielmo, Germano Mariano, Mirco Burocchi, Alessandra Campanelli, Andrea Scotti, Alessandra Pecoraro, Marco Angrisani, Paolo Ferrari, Andrea Minervini, Camilla Gasparoli, Go Wakabayashi, Giuseppe Maria Ettorre

**Affiliations:** 1Department of General, Hepatobiliary and Pancreatic Surgery, Liver Transplantation Service, San Camillo Forlanini Hospital, 00151 Rome, Italy; mcolasanti@scamilloforlanini.rm.it (M.C.); roberto.meniconi@gmail.com (R.L.M.); ferrettistefano@hotmail.com (S.F.); nicola.guglielmo@libero.it (N.G.); germachir@gmail.com (G.M.); mburocchi@scamilloforlanini.rm.it (M.B.); acampanelli@scamilloforalnini.rm.it (A.C.); ascotti@scamilloforlanini.rm.it (A.S.); alessandrapecoraro88@gmail.com (A.P.); marco.angrisani3@gmail.com (M.A.); paolo.ferrari.1011@gmail.com (P.F.); andreaminervini90@gmail.com (A.M.); camilla.gasparoli@gmail.com (C.G.); gmettorre@scamilloforlanini.rm.it (G.M.E.); 2Center for Advanced Treatment of HPB Diseases, Ageo Central General Hospital, Saitama 362-8588, Japan; go324@mac.com

**Keywords:** laparoscopic liver resections, indocyanine green dye, 3D reconstructions

## Abstract

Laparoscopic liver resections have gained widespread popularity among hepatobiliary surgeons and is nowadays performed for both standard and more complex hepatectomies. Given the increased technical challenges, preoperative planning and intraoperative guidance is pivotal in laparoscopic surgery to safely carry out complex and oncologically safe hepatectomies. Modern tools can help both preoperatively and intraoperatively and allow surgeons to perform more precise hepatectomies. Preoperative 3D reconstructions and printing as well as augmented reality can increase the knowledge of the specific anatomy of the case and therefore plan the surgery accordingly and tailor the procedure on the patient. Furthermore, the indocyanine green retention dye is an increasingly used tool that can nowadays improve the precision during laparoscopic hepatectomies, especially when considering anatomical resection. The use of preoperative modern imaging and intraoperative indocyanine green dye are key to successfully perform complex hepatectomies such as laparoscopic parenchymal sparing liver resections. In this narrative review, we discuss the aspects of preoperative and intraoperative tools that are nowadays increasingly used in experienced hepatobiliary centers.

## 1. Introduction

Liver resection is recognized worldwide as a potentially curative treatment for patients with primary and secondary malignancies and resectable disease [1,2]. Depending on the histology, the tumor biology and the extent of the disease, different types of surgical procedures can be performed, combined or not with systemic chemotherapy. Metastases from colorectal cancer (CRLM) are the most frequent secondary liver disease and most commonly require a combination of chemotherapy and partial hepatectomies to spare as much liver parenchyma as possible and allow for subsequent surgery in case of recurrence. Hepatocellular carcinoma (HCC) is the most common primary liver tumor: patients generally present with underlying liver disease from chronic conditions (viral, alcoholic, metabolic…) and when scheduled for surgery, these require special considerations in the preoperative management to reduce the chance of morbidity and mortality [3,4]. Given the ongoing debate in the literature, patients with HCC both undergo anatomical or non-anatomical liver resections depending on centers and geographic areas of treatment, with different preoperative and surgical implications [5]. On the contrary, cholangiocarcinomas both intra- and extrahepatic, generally require liver resections associated with lymphadenectomy as these tumors has been associated with a higher chance of spreading to regional lymph node stations [6,7]. While intrahepatic cholangiocarcinomas normally require partial or anatomical resections to remove as much liver parenchyma as required to achieve a negative margin, patients with perihilar cholangiocarcinomas are generally scheduled for more extensive resections (i.e., major hepatectomies with bile ducts resection and reconstructions). Whatever tumor the patient is diagnosed with and whatever surgery he or she is scheduled to undergo, a preoperative careful evaluation of the patients’ general conditions, the disease extent and the anatomy of the liver are key to plan the correct surgery for the specific patient and disease. Developments in preoperative patients’ diagnostics and intraoperative imaging, as well as recent progresses in the understanding of tumor biology and a better knowledge of the anatomy of the liver, has prompted a more advanced and precise surgical attitude to patients with liver malignancies, eventually allowing to perform more complex and tailored liver resections [8,9,10,11,12].

Despite the initial skepticism, laparoscopic liver resections (LLRs) has gained widespread popularity among hepatobiliary surgeons and is nowadays performed for both standard and more complex hepatectomies [13,14,15,16,17,18,19,20]. Given the lack of tactile sensation, the challenges with manipulations of the liver and the technical difficulties of performing intraoperative laparoscopic ultrasound, preoperative planning and intraoperative guidance is even more important in laparoscopy than in open surgery to safely carry out complex and oncologically safe hepatectomies.

In this brief review, we discuss the use of preoperative and intraoperative tools that allow to perform complex and precise laparoscopic liver resections. We reviewed the literature searching on the major scientific databases including PubMed, Scopus and Embase. The following key words were used in different combinations depending on the database: Liver surgery, hepatectomy, laparoscopy, minimally invasive, preoperative 3D imaging, 3D reconstructions, 3D printing, intraoperative navigation, intraoperative mapping, ICG, indocyanine green dye, fluorescence. Reference cross checking was also performed.

## 2. Preoperative Imaging

A safe and successful liver resection should be oncologically radical, respect the anatomy, and leave as much healthy parenchyma as possible with adequate inflow, outflow, and bile drainage. These principles are important to reduce the possibility of tumor recurrence on one hand, and decrease the chance of post hepatectomy liver failure (PHLF) on the other hand [21]. Because this balance in liver surgery is very important, the definition of the anatomy of the organ including major vascular structures and bile ducts and the relationship of these structures with the disease, requires dedicated and precise studies. With recent advancements in technology, routine imaging such as CT scans, MRIs and PET/CTs have been coupled with more advanced imaging techniques that serve as a better tool to preoperatively study the anatomy of the patient and the extent of the disease to eventually plan the surgery accordingly. Indeed, the 2D pictures obtained by standard diagnostics can be further processed to obtain 3-dimensional imaging that can further increase the knowledge of the anatomy and more accurately depict the vascular and biliary structures [22]. This allows the surgeon to tailor the procedure on the specific disease and anatomy, and therefore on the patient. Indeed, 3D modeling contributes to increase the precision of liver surgery by allowing the estimation of the total liver volume and of the single portal territories, the identification of the lesions and the relationship with the major structures and the delineation of resection lines along liver segments, eventually allowing for a preoperative simulation of the planned resection (Figure 1). Many different medical softwares are available on the market to render 3D images from conventional 2D CTs or MRIs (Syngo.via Liver Analysis, Slicer, Zionstation, Osirix, MeVis…). All of these produce a detailed reconstruction that can be used to plan many different types of procedures, from wedges and anatomical segmentectomies to major hepatectomies with or without bile duct and vascular reconstructions. In hilar cholangiocarcinoma for example, 3D imaging allows to spatially recognize the exact location of the tumor and the extent of resection, together with the exact point of transection and reconstruction of the biliary continuity to have a oncologically free margin, anticipating the type of surgery [23]. Operations performed with 3D modeling showed significantly shorter operative time, lower blood loss and fewer margin positivity [24].

A further interesting application of imaging technology in liver surgery is the use of holograms retrieved from conventional preoperative 2D imaging. Despite being still in the investigation phase, an increasing number of hepatobiliary centers are implementing the use of holograms in their practice, with interesting results both in the preoperative surgical planning and in the operating room [25]. The use of augmented reality and multiplanar reconstruction for imaging has shown similar accuracy of lesion localization but with a significant shorter time of identification compared to conventional MRIs [26].

3D printing models have been recently described as useful tools in living donor hepatectomies [27]. In oncological liver surgery, conventional 2D imaging is used to print liver casts that have all the desired information to evaluate the disease of the patients and plan the surgery accordingly. Depending on the printing material used, models can have different unique characteristics such as elasticity and softness. The important feature is that the surgeon can visualize the anatomy and the tumor trough the transparent parenchyma from any angle, manipulating the cast and designing the procedure. Drawbacks of this recent technology are the high cost of each model and the relatively long time of processing and printing that is more than 10 days [23].

All the above-mentioned technologies can be taken in the operative room and used to guide the resection and to correct the task depending on the intraoperative findings. Indeed, a screen during laparoscopic liver resections can be dedicated to the 3D reconstructions to constantly review the images. Keeping track of every single step allows to roadmap the procedure, decreasing unexpected events and minimizing futile maneuvers. Furthermore, the latest hologram technology can eliminate the need for dedicated screens in the operative room, allowing the direct interaction of the surgeon with the model thanks to the Microsoft HoloLens technology, using glasses and enhancing virtual reality [28,29].

## 3. Indocyanine Green Dye Fluorescence

Indocyanine green (ICG) is a sterile, anionic, water-soluble but relatively hydrophobic, tricarbocyanine molecule with a molecular mass of 776 Daltons [30]. Following intravenous injection, ICG is rapidly bound to plasma proteins and rapidly extracted by the liver without modifications and nearly exclusively excreted by the liver appearing unconjugated in the bile about 8 min after injection, depending on liver vascularization and function [31,32] ICG becomes fluorescent once excited using near-infra-red (NIR) light and the fluorescence released by ICG can be detected using specifically designated scopes and camera. Different surgical specialties have implemented the use of ICG in clinical practice. Indeed, fluorescence imaging can be of great help in the intraoperative visualization of anatomical structures such as lymphatics and lymph nodes, vessels, parenchyma such as the kidney and hollow viscus’ vascularization. Therefore, it is now widely accepted as an innovative and useful tool in many different surgical specialties such as ophthalmology, neurosurgery, pediatric, colorectal, gastric, and esophageal surgery [31,33,34,35].

Given its properties, ICG has numerous applications in liver surgery. First and foremost, the biliary secretion of the dye allows to visualize the bile ducts during cholecystectomies, donor hepatectomies or liver resections with bile duct resections, allowing to precisely identify the structures that need to be saved or cut. In a recent consensus conference on the applications of ICG in hepatobiliary surgery, experts found that the use of this technology, significantly improved the accuracy of bile duct identification and reduced the incidence of iatrogenic injuries [36]. Furthermore, ICG can also be used to detect bile leaks from the cutting surface, although there are only preliminary reports on the usefulness of this technology in this setting [37].

Administered 1–2 weeks before the surgery, ICG nicely helps to intraoperatively identify the tumor, either HCC, cholangiocarcinoma or CRLM. The dosage and timing of administration of the dye varies according to the disease and type of parenchyma: the general principle is that for normal liver parenchyma, a dose of 0.3–0.5 mg/Kg is administered 10 days before surgery. In cirrhotic or fibrotic livers, the dose should be decreased given the impaired washout properties of the parenchyma and should therefore not exceed 0.3 mg/Kg. Also, the timing of administration can be modified increasing or decreasing the dose of ICG whenever the surgery is earlier or later that 10 days from the infusion respectively. The dye is accumulated and then progressively washed out by the normal liver parenchyma, while it is retained by the tumor, given the altered anatomy causing distortion of the peripheral bile structures. Ishizawa et al. previously described that tumor fluorescence also varies depending on histological differentiation of tumors, with different patterns and accumulation timing [31,32].

ICG can also be administered intraoperatively to identify the portal territories and therefore perform anatomical liver resections [38]. Glissonian pedicles of first, second or third order, can be isolated either from the liver hilum or from the parenchyma. Then, a positive or negative staining technique is used. The positive staining technique reproduces what has been previously described by Makuuchi et al. for systematic segmentectomies [39]. The portal pedicle of the territory that needs to be resected is identified by ultrasonography and directly punctured and injected with ICG. The segment that needs to be resected will now shine green and the remaining liver will appear dark at near-infrared light. This technique is challenging in laparoscopy as requires advanced skills with intraoperative ultrasound and confidence with needle handling and targeting either trans- or intraabdominally. The negative staining technique is more frequently performed in laparoscopy as it is easier and ensures the same clear vision demarcation of the portal territories. The Glissonian pedicle is isolated from the hepatic hilum using the Glissonian approach, or transparenchymally using the transfissural approach. Glissonian pedicles of lobes, sections, segments and subsegments can be encircled from the liver hilum using the extra-fascial approach and respecting the Laennac’s capsule as described by Sugioka et al [40]. Once the pedicle is identified, it is temporarily clamped with an endoscopic bulldog and the ICG is administered intravenously. With this technique, the whole liver will shine green while the portal territory that needs to be resected will appear dark using near-infrared light camera (Figure 2) [41]. Both techniques are gaining worldwide popularity as the ICG allows to achieve a perfect anatomical resection. Indeed, it guarantees the identification of the intersegmental planes and the borders of the resection not only on the surface but also in the deep liver, removing the whole tumor-bearing area and leaving a well-vascularized parenchyma. In the era of minimally invasive liver surgery, ICG fluorescence helps identify liver segments boundaries and localize tumors, even in cirrhotic livers [42]. This technology, may help overcome the limitations of LLR (lack of tactile sensation, challenges in ultrasound) for both anatomical and non-anatomical resections. Intraoperative ICG guidance during laparoscopic hepatectomies has some important limitations. First, specific technology is needed, with dedicated screens and laparoscopic equipment, and despite it is nowadays widely available, it still represents an extra cost. Dosage and timing are very important, and a wrong administration of the dye will make the technology useless as all the liver will shine green or nothing will shine at all. Few cases of hypersensitivity to the dye have been reported but thorough investigation of the past medical history should anyhow be performed. Finally, both negative and positive staining are challenging and require specific technical skills which need to be acquired with dedication and significant learning curve.

## 4. Laparoscopic Anatomical Parenchymal Sparing Liver Resections

The above-mentioned technological advancements, allowed to push the limits in liver surgery and perform technically major surgeries that were once considered not feasible. These tools are of great help during anatomical liver resections. Anatomical resections include the removal of an area that is defined by the vascular supply of a portal pedicle (either main, sectorial, segmental or subsegmental). Anatomical resections allow to remove the tumor bearing area and this has been shown in retrospective reports to improve the oncological outcomes and to reduce local recurrence compared to non-anatomical resections in patients with HCC [43,44]. Moreover, anatomical resections allow to reduce the remnant liver ischemia that is common issue after partial resection, and this has been associated with worse long-term outcomes for both HCC and CRLM [45,46].

Anatomical studies show that the Glissonian pedicles first bifurcate into right and left hemi-livers. Then, secondary bifurcation divides the right hemi-liver into anterior and posterior sections and the left hemi-liver into medial and lateral sections; every section has further ramifications into segments and each segment has one or more independent pedicles [47,48]. These deep and distal ramifications introduce the concept of the “cone unit”, the smallest resectable anatomical part of the liver, supplied by a tertiary branch and with the base on the hepatic surface and the apex towards the hilum; [49] each segment consists of six to eight cone units, which can all be separately identified by one feeding pedicle and resected as independent anatomical subsegments [50].

This concept of a precise and deep understanding of the anatomy of the liver, paired with the oncological principle of anatomical resections and with the technological advancements, led some centers to perform what is known as laparoscopic anatomical parenchymal sparing liver resections (Figure 3) [33,41]. This technique requires a careful and detailed preoperative planning and advanced laparoscopic technical skills. Preoperative 3D reconstructions and surgical planning, intraoperative use of contrast ultrasonography, the intraoperative use of the Glissonian approach from the liver hilum and the Indocyanine Green Dye (ICG) negative staining are the basic steps to perform this technique that is hereby summarized.

All patients undergo the indocyanine green retention test (ICG-R15) two weeks before surgery to assess the liver function. 0.5 mg/kg are administered allowing for a correct liver function test and for a complete washout of the dye from the healthy liver by the day of surgery. Conversely, the tumor will retain the dye and will therefore allow for its identification during surgery. The liver anatomy is then reconstructed using a dedicated 3D software to evaluate the patient’s anatomy, the relationship of the tumor with major structures and the portal territories. Furthermore, the volumetry of each Glissonian pedicle is performed to perfectly know the contribution of each territory to the total liver volume and decrease the possibility of post hepatectomy liver failure. A dedicated meeting with a specialized radiologist is then carried out to plan the resection according to the reconstructions. The resection is planned to aim to the smallest but oncologically safe anatomical tumor-bearing area respecting the “cone unit” principle. The specialized radiologist returns a rendered simulation of the surgery, including the 3D mapping of the Glissonian pedicle to tackle, the portal territory to remove and the borders of the resection area. During the operation, a five trocars technique is used for all type of resections, with small adjustments depending on the side of the resection. First, laparoscopic ultrasonography with 16 μL of ultrasound contrast medium (SONAZOID, Daiichi-Sankyo, Tokyo, Japan) is performed to confirm the location of the tumor and its radiological pattern. Then, near-infrared light camera is used to visualize the tumor shining green on the liver capsule. A Glissonian approach from the liver hilum is then performed aiming to the pedicle feeding the resection area that was planned during preoperative simulation. The Laennac capsule is respected using the landmarks and gates according to Sugioka et al. [51]. The whole procedure is performed under Pringle maneuver initially with one ischemic period of five minutes as preconditioning and then with regular cycles of 15-min ischemic periods and 5-min clamp-free. During the whole procedure, the 3D preoperative simulation is checked multiple times on a dedicated screen using it as a roadmap to the correct task. Once the planned Glissonian pedicle is identified, it is temporarily clamped to create an ischemic area on the liver surface. IOUS and Doppler are used to confirm that the cyanotic area corresponds to the tumor-bearing area. A 0.5 mg intravenous bolus of ICG is then administered to allow for a negative staining: this is rapidly extracted by the liver and as a result, the whole parenchyma shines green while the clamped area remains dark confirming the ischemic area and indicating the resection borders. The resection is then carried out accordingly using energy devices and bipolar forceps. During the whole parenchymal transection phase, near-infrared camera is used intermittently to check for a proper anatomical resection; indeed, the ICG staining allows to clearly visualize the borders between the normal parenchyma (shining green) and the resection area (dark) also in the deeper liver. Finally, the Glissonian pedicle that was previously clamped is now secured with a clip or a vascular stapler. A recent publication has shown that this technique is feasible in expert hands, with good postoperative results for both HCC and CRLM [41]. Long-term oncological results are awaited.

## 5. Case Presentation

We herein present a case that was recently managed at our institution, the Department of Surgery of the San Camillo Forlanini Hospital of Rome, Italy.

A 53-year-old man with previous history of alcohol-related liver cirrhosis presents to our department for routine follow-up. His comorbidities include hypertension managed with oral antihypertensive drugs and diabetes mellitus type 2. He has no significant allergies and never underwent any surgical procedure. He brings an ultrasound, which shows a 4 cm heterogenous mass in segment 8. His alfafetoprotein level is elevated to 76 ng/mL. He has no symptoms and looks in good performance status. We scheduled him for a triphasic CT scan, which shows a lesion of 4.3 cm with brisk arterial contrast and venous washout. According to the LIRADS classification, this lesion could be considered a class 5 with diagnostic features of hepatocellular carcinoma. The patient was discussed in our multidisciplinary tumor board including hepatobiliary and transplant surgeons, hepatologists, radiologists, pathologists, oncologists, and interventional radiologists. The plan was to submit the patient to curative intent treatments given his early presentation according to the Barcelona Clinic Liver Cancer Staging System (BCLC), namely surgical resection or liver transplantation; radiofrequency ablation was excluded given the tumor’s dimensions. Given the good performance status, the position of the lesion (which was right below the Glissonian capsule) and the liver function of the patients, the MDT decided to schedule the patient for surgery. We therefore saw the patient in clinic and discussed the procedure. Informed consent was signed, and liver function was tested using ICG retention rate. We used 0.5 mg/Kg corresponding to 40 mg in this 80 kg patient. The DICOM data of the CT scan of the patient were then submitted to our radiologist who performed a 3D reconstruction of the patient’s anatomy and the relationship of the lesion with the major vessels. Furthermore, the exact dimensions of the portal territories for segment 8 were reconstructed and showed on the model. We normally aim at the narrowest but still oncologically safe resection possible. The surgery is then planned on the model, identifying the borders of the resection and the exact location of the Glissonian pedicle to tackle and the hepatic veins to skeletonize and cut. Once the preoperative surgical plan is discussed between the surgeons and the radiologists, the patient can be scheduled for surgery. This generally happens 2 weeks from the administration of ICG to achieve a complete washout of the dye by the normal parenchyma and a retention by the tumor that will then be showed intraoperatively using the narrow band camera. The patient is scheduled for a laparoscopic anatomical segment 8 resection. In our experience, we use the so called “French position” to operate laparoscopic cases, with the patient standing in between the legs of the patient and two assistants on each side. Two screens in the operating room are dedicated to the endoscopic vision, one screen is dedicated to the intraoperative ultrasound, while one dedicated screen allows to show the preoperative surgical planning and therefore guide the resection throughout the case. We use a five-trocar technique with 1 umbilical port and 4 ports on the subcostal line. One port is epigastric and is very important for the dissection of the hepatocaval confluence. An extra 5 mm access is used to perform an extracorporeal Pringle manuever. Open laparoscopy access is gained at the level of the umbilicus. After inserting all the trocars, the narrow band camera is used to identify the HCC on the hepatic dome on segment 8, which is shining green because of the ICG administered 2 weeks before. Intraoperative ultrasound and doppler are then performed to confirm the border of the resection. Pringle manuever is prepared. Dissection is started from the hepatocaval confluence to immediately identify the middle and right hepatic veins. For segment 8 resections, no extensive right lobe mobilization is necessary unless exposure is limited. We then start our parenchymal transection using a combination of energy-based device clamp-crushing technique, CUSA dissection and bipolar coagulation. We identify the middle hepatic vein at its origin, and we carry our parenchymal transection in a cranio-caudal fashion, sweeping the liver parenchyma from the vein. This avoids any tearing on small peripheral branches of the middle hepatic vein. Slowly progressing caudally, we encroach the Glissonian pedicle for segment 8, going to the lesion and vascularizing the tumor bearing area. We test the pedicle using a bulldog clamp and checking with the doppler the absence of flow in segment 8 and the presence of flow in the remnant liver. We then ask the anesthesiologist to inject 1 mL of ICG intravenously. We will then see all the liver shining green but not segment 8, which is our resection area. Guided by the ICG we will then carry out the anatomical resection. The Glissonian pedicle is stapled, and the resection is carried out dissecting the whole resection area from the middle and right hepatic veins. Once the resection is finished, the vascularization of the remnant liver is checked both with the ICG and the doppler. A drain is generally not placed unless there are specific issues during the procedure. The patient was placed on a fast-track protocol with early feeding and mobilization and was discharged home on postoperative day 4. Follow up is now more than 1 year and the patient is currently in good health status with no signs of recurrence.

## 6. Conclusions

Technological advancements in liver surgery have allowed to increase the preoperative knowledge of the patient’s anatomy and plan the procedure accordingly. Preoperative 3D imaging and reconstructions, 3D printing and ICG guided surgery are tools that are increasingly used in experienced hepatobiliary centers. In modern laparoscopic liver surgery, these technological improvements allow to tailor the procedure to the specific patient and disease, minimizing the chance of postoperative events meanwhile ensuring a safe and oncologically sound resection.

## Figures and Tables

**Figure 1 diagnostics-11-02169-f001:**
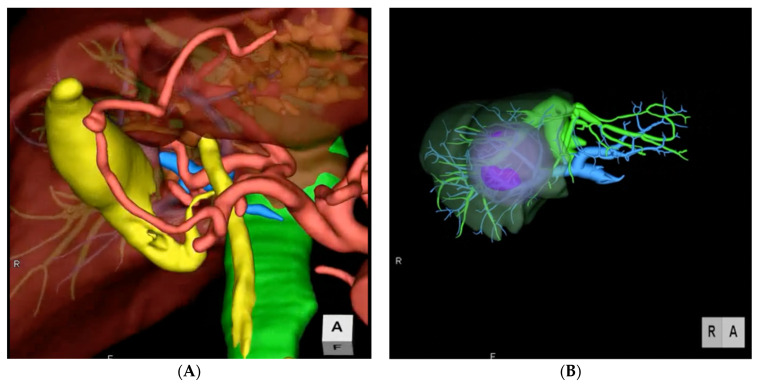
Two examples of preoperative 3D imaging. (**A**) Reconstruction of arterial, portal, and venous system and biliary tree. (**B**) Reconstruction of vascular structures, portal territories and tumor.

**Figure 2 diagnostics-11-02169-f002:**
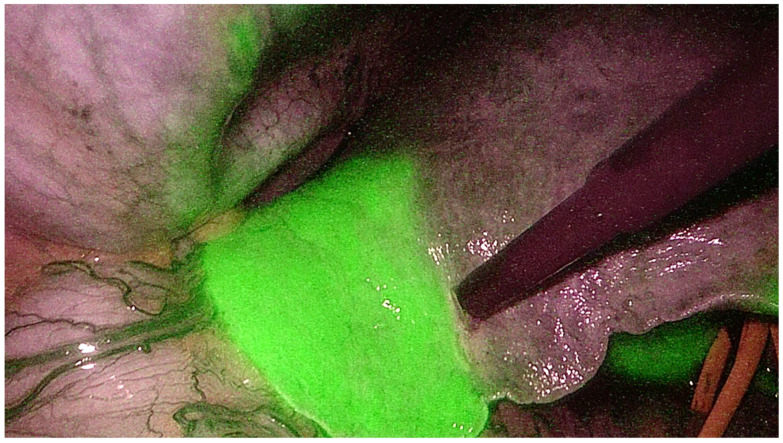
Indocyanine green dye negative staining technique to identify the portal territories during anatomical hepatectomy.

**Figure 3 diagnostics-11-02169-f003:**
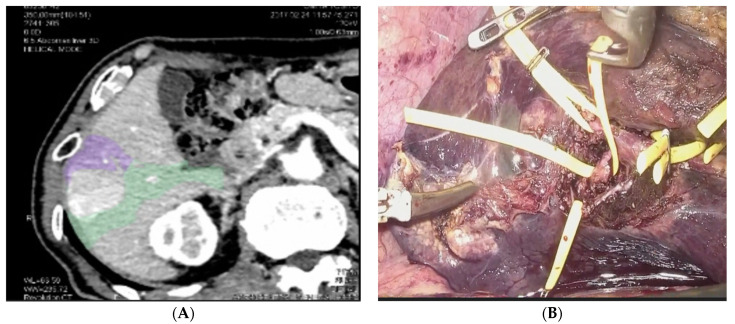
Anatomical Parenchymal Sparing Liver Resections: (**A**) 3D reconstruction with tumor bearing area from pedicles of segment 5 and segment 6. (**B**) Intraoperative view of the isolation of the Glissonian pedicles for segment 5 and 6.

## Data Availability

Not applicable.

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
