# Peer review of "The Applications of 3D Imaging and Indocyanine Green Dye Fluorescence in Laparoscopic Liver Surgery"

_diagnostics, 2021, doi:10.3390/diagnostics11122169_

Round 1
Reviewer 1 Report
This is well organized and concise review.
Timing and dose of ICG are known to be various according to indications.
It will be better that you will suggest and summarize timing and dose of ICG according to indications.
Author Response
We would like to thank the reviewer for taking time to review our manuscript.
We agree that timing and dosage of ICG varies according to the disease and the type of parenchyma. We have further addressed this issue in the manuscript.
We hope that this answer your question.
Sincerely,
Dr. Giammauro Berardi MD, PhD, FEBS (HPB)
Reviewer 2 Report
The current study aims to discuss the aspects of preoperative and intraoperative tools in hepatobiliary surgery using Indocyanine green dye.
The authors reviewed the current literature.
The study demonstrates that these technological improvements allow to tailor the procedure to the specific patient and disease, minimizing the chance of postoperative events meanwhile ensuring a safe and oncologically sound resection.
The authors should be congratulated for the work and for addressing an important topic. Only few points warrant mentions:
Major comment:
- In “Introduction” section, authors should declare how the narrative review of the literature was carried out;
- In “Indocyanine Green Dye Fluorescence” section, authors should show the limits of the technique.
- To give a comprehension of the importance that ICG is gaining in surgical procedure, I suggest to show more applications as shown in: - Esposito C, Coppola V, Del Conte F, et al. Near-Infrared fluorescence imaging using indocyanine green (ICG): Emerging applications in pediatric urology. J Pediatr Urol. 2020;16(5):700-707. doi:10.1016/j.jpurol.2020.07.008; and - Esposito C, Settimi A, Del Conte F, et al. Image-Guided Pediatric Surgery Using Indocyanine Green (ICG) Fluorescence in Laparoscopic and Robotic Surgery. Front Pediatr. 2020;8:314. Published 2020 Jun 17. doi:10.3389/fped.2020.00314
Minor comments:
- In “Indocyanine Green Dye Fluorescence” section, authors should mention how much ICG must inject.
Author Response
Thanks for taking time to review our manuscript, and thanks for your interesting comments that cleary improve the quality of our manuscript.
Replying to your points below:
- We have clarified in the introduction that this is a narrative review and that we conducted the research using Pubmed, Embase and Scopus using specific terms and using cross-check references (lines 65-71)
- Thanks for your correct point. We have now addressed the limitations of the technique (line 194-203)
- Thanks for your suggestion, we have added some ideas to this and the references your pointed out (lines 138-144)
- We have added clear dosages and timing of infusion for different disease (lines 155-162).
Thanks again for your nice comments.
Best,
Giammauro Berardi MD, PhD, FEBS (HPB)
Round 2
Reviewer 2 Report
It's ok.